# Perceived paternal emotional fertility intention and its correlates in Ethiopia among a cohort of pregnant women: Community based longitudinal survey; A secondary data analysis of the 2019/20 baseline survey

Solomon Abrha Damtew[1]*, Mahari Yihdego Gidey[2], Niguse Tadele Atnafu[3], Fitsum Tariku Fantaye[4], Kelemua Mengesha Sene[5], Bezawork Ayele Kassa[6], Hailay Gebremichael Gebrekidan[7], Tariku Tesfaye Bekuma[8], Seifu Yenneda Berhe[9], Gelane Duguma Edosa[10], Temesgen Bati Gelgelu[10], Wakgari Binu Daga[11], Dereje Haile[10], Tesfamichael Awoke Sisay[2], Ayanaw Amogne[2], Tariku Dejene Demissie[12], Assefa Seme[13], Solomon Shiferaw[13]

1 Department of Epidemiology and Biostatistics, School of Public Health, Wolaita Sodo University, Wolaita Sodo, South Ethiopia, Ethiopia, 2 PMA Ethiopia Project and related projects at Addis Ababa University, Addis Ababa, Ethiopia, 3 School of Nursing and Midwifery, Addis Ababa University, Addis Ababa, Ethiopia, 4 FTF Research consult, Addis Ababa, Ethiopia, 5 Department of English Language, Kotebe Metropolitan University, Addis Ababa Ethiopia, 6 University of Cape Town, Cape Town, South Africa, 7 School of Public Health, College of Health Sciences, Mekelle University, Mekelle, Tigray, Ethiopia, 8 Department of Public Health, Institute of Health Sciences, Wollega University, Nekemte, Ethiopia, 9 Ethiopian Statistical Services, Addis Ababa, Ethiopia, 10 School of public health, Wolaita Sodo University, Wolaita Sodo, Ethiopia, 11 Department of Public Health, Ambo University, Ambo, Ethiopia, 12 Center for Population Studies, College of Developmental Studies, Addis Ababa University, Addis Ababa Ethiopia, 13 School of Public Health, Addis Ababa University, Addis Ababa, Ethiopia,

* solomon.abrha@wsu.edu.et

## Abstract

### Background

Perceived paternal emotional fertility intentions were measured by asking pregnant women how their husbands felt when they have learnt about the index pregnancy. Paternal emotion during pregnancy and childbirth is imperative for better maternal and newborn health outcomes, though policy and strategic framework has been lacking in Ethiopia. Therefore, this study aimed to explore perceived paternal emotional fertility intentions of their husbands and/or their male partners and examine its correlates among a panel of pregnant women in Ethiopia.

### Methods

Nationally representative data from cohort one baseline cross-sectional survey were used. A total of 2,115 pregnant women from a total of 217 enumeration areas were included in this further analysis. Frequencies were computed to describe pregnant women.

**Data availability statement:** The datasets generated during the study are publicly available from the PMA website and/or the Johns Hopkins Research Data Repository: https://archive.data.jhu.edu/dataset.xhtml?persistentId=doi:10.34976/h75w-8084

**Funding:** The author(s) received no specific funding for this work.

**Competing interests:** The authors have declared that no competing interests exist.

**Abbreviations:** AOR, Adjusted Odds Ratio; CS, Cross Sectional; EA, Enumeration Areas; HH, Households; PMA, Performance for Monitoring for Action Ethiopia Service Delivery Point; SNNPR, The former Southern Nations, nationalities and Peoples Region; US, United States of America.

Multinomial logistics regression statistical modeling was fitted to identify correlates of perceived paternal emotional fertility intentions. Results were presented in the form of percentages and odds ratio with 95% Confidence Intervals. Statistical significance was declared at a p-value of 0.05.

## Result

The proportion of perceived paternal emotional fertility intentions of being a sort of happy and very happy were found to be (35.40%; 95%CI: 33.00%, 37.87%) and (49.03%; 95%CI: 46.48%, 51.6%) respectively. The likelihood of perceived paternal emotional fertility intentions of being very happy was (AOR: 95%CI: 5.06: (1.73, 14.85) and (AOR: 95%CI: 2.65: (1.67, 4.20) times higher among older pregnant women and those who intended having another child respectively. On the contrary, those with higher birth order, who wanted no more another child, those living as a partner and; those residing in Addis Ababa and SNNPR had (AOR: 95%CI: 0.25: (0.15, 0.40), AOR: 95%CI: 0.14: (0.07, 0.27); (AOR: 95%CI: 0.54: (0.33, 0.90), (AOR: 95%CI: 0.34: (0.17, 0.67), (AOR: 95%CI: 0.27: (0.14, 0.53) and AOR: 95%CI: 0.25: (0.15, 0.40) times lower likelihood of perceived paternal emotional fertility intention of being very happy about the index pregnancy respectively. The likelihood of perceived paternal emotional fertility intentions of being a sort of happy was found to be (AOR: 95%CI: 1.93 (1.21, 3.10) times higher among those wanting to have another child. This likelihood was found to be (AOR: 95%CI: 0.62 (0.43, 0.89), (AOR: 95%CI: 0.43 (0.22, 0.85) and (AOR: 96%CI: 0.45 (0.28, 0.74) times lower among those whose desired place of delivery was health facility, with higher birth order and residents of Oromiya region respectively.

## Conclusion

Half of the pregnant women perceived that their husband felt very happy with calls up on a region specific age sensitive interventions in improving couples communication, and discussion on the spacing and timing of pregnancies as well as to work on improving childbirth preparedness and complication readiness. The ministry and relevant partners need to work strategically on male's involvement in fertility desire along with emotional care and support. Women with future fertility intention, lower birth orders and those who have not legally married need to be targeted. Installing inter pregnancy preconception care package; improving counseling and provision of postpartum contraceptives; increasing men fertility knowledge and their emotional readiness; and lifestyle adjustment before pregnancy to improve psycho-social health during the perinatal period and paternal emotional fertility intentions are imperative.

## Background

The Ethiopian population is one of the largest in Africa. In consideration with this numerous activities have been conducted to reduce the average number of births per woman by the government of Ethiopia and the Health Minister. The aim is to make parallel the population growth with the economic development through the provision of family planning counseling and actual provision of services [1–3]. However, it has not been feasible to achieve the desired level of change as was planned and intended in the national health sector transformation plan

(HSTP) and reproductive health (RH) strategies [4–6]. Hence, the annual population growth and fertility rate remain at a high rate of 2.7 and 4.6, respectively [7].

The absence of paternal emotional support during pregnancy and lack of couple's psychological readiness towards fertility and child bearing have contributed for this sustained higher fertility rate and annual population growth which subsequently led to negative health and economic consequences [8,9] from the family to country level. Evidences shown that education and family planning programs could play an imperative role in fertility desire trends, influencing emotional fertility intention [8]. Spéder & Kapitány stated that happier men and women prefer to become parents sooner arguing the link between intention and behavior is stronger [10,11]. Moreover, couples´ happiness and emotion determine having a child: particularly women's happiness and their perceived paternal emotional fertility intention matters more in decision making towards having a subsequent child [12]. Another evidence showed that fertility intention and behavior have a closer link [13] indicating emotional fertility intentions and behavior influence the demographic dividend of countries. However, there is a dearth of evidence on the emotional aspect of fertility intention among couples in general and husbands in particular in Ethiopia and other low and middle income countries.

This has been further precipitated by the lack of a policy framework for males' involvement during pregnancy, childbirth and towards emotional fertility intentions in low and middle income countries. Such a policy when exists a gap in policy and practice has been witnessed in developing countries context [14,15]. Articulating and endorsing such a policy is likely to mitigate males' dominance on emotional fertility desire and enhance their emotional care and support during the perinatal period. Extending such a policy framework to paternal fertility emotion in particular and decision making as far as sexual and reproductive health is concerned could have paramount importance for improved maternal and newborn outcomes [16–19]. Further evidence also showed that considering discordance in fertility desire among couples which has resulted from the sole males decision making power over family size [20], including dominance in reproductive health service use [21,22] has been imperative for considerable males involvement in emotional care and support during pregnancy and childbirth as far as fertility is concerned.

Moreover, fertility intentions are an integral part of reproductive health (RH) right which can be considered as decision making power over their fertility, family wellbeing and the country's population demographic composition dividend [23–25]. However, in low and middle income countries including Ethiopia where males dominance is culturally constructed and socially accepted, males took the lead in every decision making process for the family ranging from household level decision making to limiting family size and reproductive health services use. In such a context, women may not have their voices heard. In low and middle income countries fertility desire is determined by the husband and/or partner as well [16]. This is the reflection of that the husbands´ emotional reaction towards pregnancy and child birth are critical in determining the women emotional feeling and wellbeing during their pregnancies and in improving maternal and newborn outcomes [15]. Hence, due to such husband dominance on fertility desire; women are likely to held negative and unpleasant feelings and emotion whenever they thought of pregnancy and childbirth [18,26]. Due to this they also tend to held negative emotion for every additional child they are going to bear [10,16,27].

To this end, studies are skewed in measuring women´s emotional fertility intentions [26] and their fertility desire [18,28–32]. Paternal emotional aspect of fertility intentions and how wives' would perceive that their husbands and/or partners when learnt about the index pregnancy is less explored and there is a dearth of evidence in Ethiopia. Such paternal emotion about the index pregnancy might be related with religion and cultural acceptability of large families since husbands play a pivotal role in determining the number of children that

the family should have [14,33–35]. Besides, as per the World Health Organization definition of health, this study attempted to address the mental and/or emotional and social aspect of health since pregnancies and childbirth are communal events where husbands are largely involved in Ethiopian and in other low and middle income countries [36].Therefore, generating evidence on the level of perceived, paternal roles, expectations, experiences and challenges faced by men who wish to be involved in maternal health issues, particularly during pregnancy and childbirth is very critical to improve maternal and fetal outcomes in the perinatal period [14,15]. This is critical so that husbands could be emotionally stable and being supportive [37] when learned their wives´ pregnancies so as to be able to provide holistic emotional support and care during their wives' pregnancy, childbirth and in the post-partum period. Hence, determining the level of perceived paternal emotional fertility intentions about the index pregnancy among a cohort of pregnant women and identifying its correlates contributing for such variation is very critical for improving newborn and maternal health outcomes. This could provide actionable evidence for the Federal Democratic Republic of Ethiopian Health Minister and developmental partners. Therefore, this study aimed to explore the perceived emotional fertility intentions of male partners and to examine its correlates among a cohort of pregnant women in Ethiopia.

## Methods and data sources

### Data sources, study design, population and sample size

Performance monitoring for action Ethiopia have been collecting both cross-sectional and longitudinal data on selected maternal, newborn health and contraceptive use and women and girls empowerment indicators over the past decade.

This study used community based baseline cross-sectional data from prospective cohort study with 6 weeks, 6 months and one year postpartum follow up interviews apart from the baseline cross sectional data used for this study. The data were collected from pregnant women in Ethiopia from six regions: namely: Addis Ababa, Afar, Amhara, Oromia, SNNPR and Tigray by well experienced resident enumerators. The study was started by recruiting and enrolling pregnant women and puerperal women less than six weeks which was then followed by administering the female baseline questionnaire. Then these panel of women were interviewed at 6 weeks, 6 months and one year postpartum as follow up interviews. However, this study further analyzed and present data from the baseline cross-sectional survey among the pregnant women during the enrollment.

Two thousand two hundred thirty nine (2239) pregnant women provided response for the question "When your partner found out you were pregnant, how did he feel?" One hundred twenty four (124) were exclude from the analysis due to: 3 women provided no response, 46 reported do not know, 57 reported that they have not told to their partner and 18 of them reported they have no partner. Hence, the final analytical sample size was restricted to 2,115 pregnant women.

Performance Monitoring for Action (PMA) employed a two stage cluster sampling; in the first stage enumeration areas (EA) were selected. In each of the selected EAs census was conducted to screen and enroll pregnant women and women less than 6 weeks postpartum women by then. This study was restricted to only 2,115 pregnant women enrolled from 217 enumerations areas and who completed the baseline female questionnaire. Following enrollment the female baseline questionnaire was administered. In the female baseline questionnaire women were asked about their antenatal care sought thus far during their index pregnancy, partner support and perceived community encouragement on the use the three main domains of the maternal and newborn care components; their reproductive and sexual history, their

birth preparedness and complication readiness; about their agreement and disagreement on girls and women empowerment towards contraceptive use and women sexual and reproductive issues; about their contraceptive use history and their current and future fertility intention. Women were asked how they themselves did felt when learned the index pregnancy.

Most important to this study was that they were also asked how their husband felt when their husbands and/or partners have learnt about the index pregnancy. This variable was regarded as women perceived paternal emotional fertility intentions towards the index pregnancy which is the main outcome variable of this study. The analytic sample for the multinomial logistics regression was restricted to pregnant women who are married and current living together with a partner so as to measure those actively in cohabitation currently. The details on the indicators measured, sampling procedure, field operation and quality assurance techniques employed by the project runners were well described in the published protocol [38].

In order not to miss all pregnant and six week postpartum women by then in the selected enumerations areas (EAs), complete census was conducted in each of the selected enumeration areas. The main sample units or enumeration areas (EAs) were chosen using the frame from the Ethiopia Population and Housing Census (PHC), which was performed in 2019 by the Ethiopia Central Statistical Services. Of the total 265 EAs which were chosen in the first stage with independent selection in each sampling stratum and a probability proportional to EA size; pregnant women from 217 EAs which were selected from the six panel regions were included in this further analysis. In second stage, a census of all households was conducted in all the selected EAs to obtain adequate sample size of pregnant women and to improve the study´s power. The protocol of PMA Ethiopia contains all the details on sample design and selection probabilities, design effects and sampling methods. More detail in the details on sampling design and selection procedures and field work implementation were described well in the protocol [39].

The cross sectional baseline data for cohort one were collected from November 2019 to January 2020. The details in field work implementation were reported elsewhere while the follow up extended up to Jul 2021 [39]. The first field work for the panel baseline survey was commenced by screening and enrolling pregnant women and women who were less than 6 weeks post-partum. The Ethiopian Central Statistical Services head office was involved in the sampling of the primary sampling unities (enumeration areas) and facilitated selected enumeration area maps. The main sample units or enumeration areas (EAs) were chosen using the frame of the Ethiopia Population and Housing Census (PHC), which was performed in 2019 by the Ethiopia Statistical Services. This panel survey was executed by Addis Ababa University's School of Public Health in collaborative efforts with the Ethiopian Public Health Association with assistance from the Federal Health Ministry, Ethiopia Statistical Services, Bill & Melinda Gates Institute for Population and Reproductive Health (Johns Hopkins Bloomberg School of Public Health) [38].

### Variables

**Outcome variable.**  The main outcome variable was pregnant women perceived paternal emotional fertility intention about the index pregnancy among a cohort of pregnant women. Pregnant women were asked single Likert scale question "when you found out you were pregnant how did your husband and/or partner feel?" This was used to measure pregnant women perceived paternal fertility emotion about the index pregnancy when their husbands and/or partner had learnt the index pregnancy among pregnant women (Table 1).

### Independent variables

Independent variables were classified into individual-level variables and enumeration area-level variables. Individual-level independent variables further categorized into

**Table 1. Items used to measure perceived paternal emotional fertility intentions among a panel of pregnant women, evidence from cohort one baseline data, 2019 to 2020.**

| Index pregnancy emotional feeling Women were asked When you found out you were pregnant how did your husband feel? | Variable | Question Items and Responses | Recode Categories |
|---|---|---|---|
| | Your Husband feel when he learned the index pregnancy | 1. Very Happy | 1. Very unhappy |
| | | 2. Sort of happy | 2. Sort of unhappy 3. Mixed Happy and Unhappy 4. Sort of Happy 5. Very Happy |
| | | 3. Mixed happy and unhappy | Other categories, such as DNK, NR, have not partner and have not told response option categories were excluded from the analysis. Responses from 124 pregnant women were exclude from the analysis due to 3 women provided no response, 46 reported no, and 57 reported that they have not told to their partner and 18 reported they have no partner. |
| | | 4. Sort of unhappy | |
| | | 5. Very unhappy | |

socio-demographic/economic characteristics variables, parity and other reproductive health (RH) characteristics and contraception use history were considered in the study.

Group or enumeration area (EA) level variables included two integral variables namely, region and place of residence which were considered. "Region" was grouped into six categories 1 = Tigray 2 = Afar 3 = Amhara, 4 = Oromia, 7 = SNNPRs and 10 = Addis Ababa city administration. The place of residence follows the default urban/rural classification. Two additional derived enumeration area level variables were computed from the individual level constructs.

The following individual and enumeration area level variables were considered in this analysis. Sociodemographic: age of women, educational status, wealth index, marital status and religion of the women; reproductive health characteristics: birth order, marriage type, marriage history, current and future fertility intention; health service use related variables such as contraceptive ever use history and planned birth attendant and planned place of delivery for the index child. The integral enumerating area level variables were region and place of residence. Note that derived enumeration area level variables were EA level wealth and proportion of women with secondary education and above were found to be multicollinear with the respective individual level constructs and hence, dropped off from this piece of particular further analysis.

The list of individual level and enumeration area level independent variables and the final recoded categories has been depicted below.

| Variable name and Categories | |
|---|---|
| Age category | 15–19 years |
| | 20–24 years |
| | 25–29 years |
| | 30–34 years |
| | 35–39 years |
| | 40–49 years |
| Educational Status | No Formal Education |
| | Primary Education |
| | Secondary Plus |

| Variable name and Categories | |
|---|---|
| Religion | Other |
| | Orthodox |
| | Protestant |
| | Muslim |
| Wealth Index | Lowest quintile |
| | Lower quintile |
| | Middle quintile |
| | Higher quintile |
| | Highest quintile |
| Parity | No Child |
| | 1_2 Children |
| | 3_12 Children |
| Marriage History | Only Once |
| | More than Once |
| Marriage Type | Monogamy |
| | Polygamy |
| Future Fertility Intention | Undecided |
| | Have a/another child |
| | No more or prefer no child |
| Marital Status | Married |
| | Living With A partner |
| | Widowed or Separated |
| Contraceptive Ever Use | No |
| | Yes |
| Residence | Urban |
| | Rural |
| Region | Tigray and Afar |
| | Amhara |
| | Oromiya |
| | SNNPR |
| | Addis Ababa |
| Desired Birth Attendant | No One |
| | Health Professional |
| | Family Member |
| Desired Delivery Place | Home |
| | Government and Private Health Facility |

## Analysis and measurement

The panel baseline cross-sectional women data set were used for this analysis. Stata software version 16 was used for this analysis. Frequencies and percentages were computed to characterize the study population. Chi-square test statistics was computed to check cell sample size adequacy and the sample size was found to be adequate to provide unbiased estimates on perceived paternal emotional fertility intentions about the index pregnancy among a cohort of pregnant women.

Exploratory data analyses were run for data cleaning thereby checking item nonresponse rate for every variable and don't know response which were later excluded from the analyses. Following this variable were recoded to create biologically plausible categories alongside

checking distribution of the recoded variables using mean and proportion. No sign of multicollinearity detected among variables in the final model except for the derived group-level variable with the respective constructs collected at the individual level.

Multinomial logistics regression statistical model building was fitted to identify important predictor's of perceived paternal emotional fertility intentions about the index pregnancy among a cohort of pregnant women. At bivariate analysis a p value cut of 0.25 [40] was used to select candidate variables for multivariate multinomial logistics regression analysis. Results were presented in the form of percentage, and odds ratio with 95% CI. Significance was declared at a significance level of 0.05. Results were reported based on weighted count.

The model fitness test was checked using the command «mlogitgof» and the result showed that the model was fit meaning that variables included in the final multinomial multivariate logistics regression model explains for the observed variation in perceived paternal emotional fertility intention about the index pregnancy among pregnant women across the categories of the independent variables with chi-squared statistic = 12.837 and Prob of chi-squared = 0.685.

## Data quality management and control

Data completeness for variables and items for creating composite variables was checked by exploratory data analysis after which any item nonresponse was excluded from the analysis. Frequency run to exclude responses with do not know (DNK) and no response (NR).

Performance Monitoring for Action Ethiopia (PMA Ethiopia) data were collected using standard and pretested tool which were translated in to three local languages (Tigrigna, Afan Oromo and Amharic) after the provision of hands-on intensive ToT and RE training with mock interviews. Besides, close supervision during filed work, timely progress report and hierarchal errors corrections were made, 10% random check with dedicated re-interviews were some of the modalities used to maintain the quality of the collected data, the detail is reported somewhere else [39].

## Ethical consideration

This study involved a secondary analysis of de_identified data from the PMA Ethiopia. The PMA Ethiopia survey was conducted strictly under the ethical rules and regulations of world health organization and IRB of Ethiopian Health and Nutrition Research Institute (EHNRI). Informed consent was obtained from respondents during the data collection process of PMA Ethiopia on data collection on Nov 2019 to Jan 2020. PMA surrey has been also conducted after obtained ethical approval from the College of Health Sciences at Addis Ababa University and Bloomberg School of Public Health at Johns Hopkins University in Baltimore, USA. PMA_ETH Publicly available Cohort one baseline and six weeks postpartum datasets were accessed after submitting a concept note for this piece of specific work form the PMA data cloud server archive via. https://www.pmadata.org/data/request-access-datasets.

Since this study involves analysis of already collected secondary data there is no need to consent, rather, concept note was submitted to get permission for data use.

"Minors less than 15 years as per the law were not included in this study. Informed verbal consent was take from study participants." Moreover, women of reproductive age group or women child bearing age were include in the study. The survey includes topics related to family planning, sexual history and other reproductive health issues which are declared as rights of women by international declarations and as supported by evidence: Rimon JGII, Tsui AO. Regaining momentum in family planning. Global Health Science Practice. 2018; 6(4):626–628. https://doi.org/10.9745/GHSP-D-18-00483. 2018 and related with waived room to directly ask

herself about her sexual and reproductive issues. Moreover standard surveys including Demographic and health surveys include women of child bearing age, as this study did.

## Result

### Magnitude of perceived paternal emotional fertility intention about the index pregnancy among a panel of pregnant women, cohort one baseline data, Nov 2019 to Jan 20

This study reported on paternal emotional fertility intention among a panel of pregnant about the index pregnancy. Responses on how pregnant women perceived paternal emotional fertility intention (on how pregnant women perceived on what their husbands and/or partners felt when learnt about the index pregnancy) by the time they were pregnant from a total of 2,115 pregnant women were further analyzed in this study.

The proportions of parental emotional fertility intentions of being felt a sort of happy and very happy about the index pregnancy among a cohort of pregnant women were found to be 1/3 (95% CI: 35.40%; 33.00%, 37.87%) and a half (95% CI: 49.03%; 46.48%, 51.6%) respectively. Moreover, closer to 1 in 6; 15.57%: (95% CI: 13.72%, 17.62%) of the pregnant women perceived that their husbands and/or partners specifically felt very unhappy/ sort of unhappy/ felt mixed feelings when learnt about the index pregnancies (2% very unhappy, 5% sort of unhappy and 8% mixed happy and unhappy feelings) (Fig 1).

### Distribution of perceived paternal emotional fertility intention about the index pregnancy among a panel of pregnant women, Nov 2019 to Jan 20

The level of perceived paternal emotional fertility intention about the index pregnancies among a panel of pregnant women when their husbands and/or partners learned about the wives´ index pregnancy has shown variation across the categories of the independent variables. The proportion of pregnant women who perceived their husbands and/or partners felt very happy, sort of happy and mixed feelings or unhappy when their husband learnt about the index pregnancy among women aged 20 to 24 years was found to be 53.03% 32.04% and 14.92% respectively. While, among the pregnant women who had attended secondary education or above, 10.6%, 26.07% and 63.30% of the pregnant women perceived that their husband and/or partners felt mixed feelings or unhappy, sort of happy and very happy when their husband learnt about the index pregnancy respectively. Similarly, this same figure stood 18.32% 39.14% and 42.54% respectively for protestant religion followers and it stood 9.8%, 32.53% and 57.64% respectively among residents of the well to do households (Table 2).

Among the pregnant women who reported higher birth order, 21.82%, 40.77% and 37.41% of them perceived that their husbands and/or partners felt mixed feelings or unhappy, sort of happy and very happy when their husband learnt about the index pregnancy respectively. This same figure stands 22.25%, 37.64% and 40.11% respectively among those who reported undecided whether to have an additional child. Similarly among women living together as a partner, 26.73%, 32.92% and 40.35% of them respectively perceived that their husbands felt mixed feelings or unhappy, sort of happy and very happy when their husband learnt about the index pregnancy respectively. Likewise, by the time their husbands and/or partners became aware of the index pregnancy, the perceived paternal emotional fertility intentions were found to be 15.68%, 35.41% and 48.92% respectively among pregnant women who reported that they ever used contraceptives (Table 2).

Among the pregnant women who reside in Tigray and Afar 7.8%, 19.91% 72.21% of them perceived that their husband and/or partner felt mixed feelings or unhappy, sort of happy and

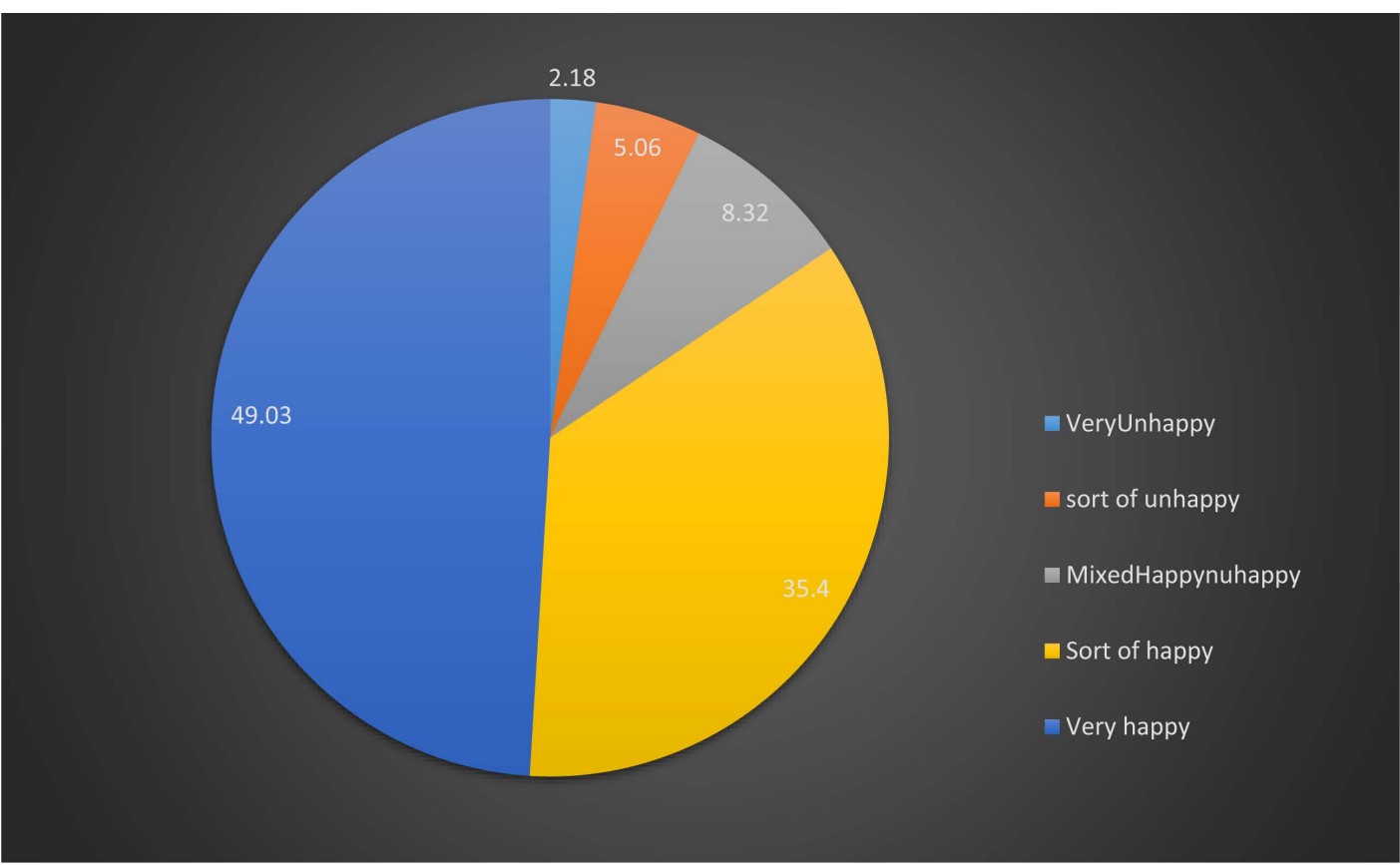

**Fig 1. Perceived paternal emotional fertility intentions about the index pregnancy among a panel of pregnant women, 2019 to 20.**

very happy respectively when they learned that their wives 'were pregnant while it was found to be 10.46%, 31.46% and 58.08% respectively for resident of Addis Ababa. This same figure stood at 17.04%, 36.71% and 46.25% among rural residents (Table 2).

### Correlates of perceived paternal emotional fertility intention about the index pregnancy among a panel of pregnant women and its correlates in Ethiopia, evidence from cohort one baseline data, Nov 2019 to Jan 20

The likelihood of perceived paternal emotional fertility intention of being very happy was found to be higher among older women and those who intended to have another child. On the other hand, those with higher birth order with 3 to 12 children, who wanted no more another child, and those living as a partner and residing all regions other as compared with Tigray and Afar had a lower likelihood of perceived paternal emotional fertility intention of being very happy about the index pregnancy among this panel of pregnant women (Table 3).

The likelihood of perceived paternal emotional fertility intention of being a sort happy about the index pregnancy among this panel of pregnant women was found to be higher among those who wanted to have another child. On the contrary, the likelihood of perceived paternal emotional fertility intentions of being a sort of happy was found lower among those with higher birth order, those who did not wanted to have an additional child, those who reside in Oromiya region and those whose desired place of delivery was health facility (Table 3).

**Table 2. Distribution of perceived paternal emotional fertility intentions about the index pregnancy among a panel of pregnant women, Nov 2019 to Jan20, N = 2115.**

| Variable | | Mixed Feelings or Unhappy | % | Sort Happy | % | Very Happy | % | Total |
|---|---|---|---|---|---|---|---|---|
| Age category | 15–19 years | 14 | 6.44 | 77 | 35.18 | 128 | 58.38 | 220 |
| | 20–24 years | 77 | 14.92 | 165 | 32.04 | 273 | 53.03 | 514 |
| | 25–29 years | 89 | 14.1 | 224 | 35.72 | 315 | 50.18 | 628 |
| | 30–34 years | 77 | 19.67 | 136 | 34.56 | 180 | 45.77 | 394 |
| | 35–39 years | 57 | 20.83 | 120 | 43.98 | 96 | 35.18 | 273 |
| | 40–49 years | 16 | 18.05 | 26 | 30.13 | 45 | 51.82 | 86 |
| | Total | 329 | 15.57 | 749 | 35.4 | 1037 | 49.03 | 2115 |
| Educational Status | No Formal Education | 152 | 17.88 | 352 | 41.42 | 346 | 40.7 | 850 |
| | Primary Education | 134 | 15.64 | 290 | 33.87 | 432 | 50.48 | 855 |
| | Secondary Plus | 44 | 10.63 | 107 | 26.07 | 259 | 63.3 | 410 |
| | Total | 329 | 15.57 | 749 | 35.4 | 1037 | 49.03 | 2115 |
| Religion | Other* | 6 | 13.27 | 19 | 45.2 | 18 | 41.53 | 42 |
| | Orthodox | 91 | 11.42 | 280 | 35.25 | 424 | 53.33 | 795 |
| | Protestant | 107 | 18.32 | 229 | 39.14 | 249 | 42.54 | 586 |
| | Muslim | 126 | 18.16 | 220 | 31.78 | 346 | 50.05 | 691 |
| | Total | 329 | 15.57 | 749 | 35.4 | 1037 | 49.03 | 2115 |
| Wealth Index | Lowest quintile | 72 | 18.13 | 165 | 41.41 | 161 | 40.45 | 399 |
| | Lower quintile | 83 | 20.11 | 148 | 36.11 | 180 | 43.77 | 410 |
| | Middle quintile | 74 | 17 | 146 | 33.39 | 217 | 49.61 | 438 |
| | Higher quintile | 56 | 13.27 | 145 | 34.11 | 224 | 52.62 | 425 |
| | Highest quintile | 44 | 9.84 | 144 | 32.53 | 255 | 57.64 | 443 |
| | Total | 329 | 15.57 | 749 | 35.4 | 1037 | 49.03 | 2115 |
| Parity | No Child | 23 | 4.41 | 144 | 27.92 | 348 | 67.67 | 515 |
| | 1_2 Children | 114 | 15.92 | 244 | 34.11 | 357 | 49.97 | 715 |
| | 3_12 Children | 192 | 21.82 | 360 | 40.77 | 330 | 37.41 | 882 |
| | Total | 329 | 15.58 | 747 | 35.38 | 1036 | 49.04 | 2112 |
| Marriage History | Only_ Once | 283 | 15.41 | 659 | 35.88 | 895 | 48.71 | 1838 |
| | More than_ Once | 44 | 16.84 | 83 | 31.49 | 136 | 51.68 | 263 |
| | Total | 328 | 15.59 | 742 | 35.33 | 1031 | 49.08 | 2101 |
| Marriage Type | Monogamy | 273 | 14.51 | 672 | 35.69 | 938 | 49.8 | 1884 |
| | Polygamy | 49 | 24.99 | 66 | 33.75 | 81 | 41.26 | 196 |
| | Total | 322 | 15.5 | 738 | 35.5 | 1019 | 49 | 2080 |
| Fertility Intension | Undecided/DKN | 41 | 22.25 | 70 | 37.64 | 75 | 40.11 | 186 |
| | Have a/another child | 170 | 11.14 | 521 | 34.16 | 834 | 54.7 | 1524 |
| | No more or prefer no child | 118 | 29.19 | 158 | 39 | 129 | 31.81 | 405 |
| | Total | 329 | 15.57 | 749 | 35.4 | 1037 | 49.03 | 2115 |
| Marital Status | Married | 307 | 15.13 | 722 | 35.56 | 1001 | 49.31 | 2030 |
| | Living With A partner | 17 | 26.73 | 21 | 32.92 | 26 | 40.35 | 65 |
| | Widowed/Separated | 5 | 27.58 | 6 | 32.22 | 7 | 40.2 | 18 |
| | Total | 330 | 15.6 | 749 | 35.45 | 1034 | 48.95 | 2113 |
| Contraceptive Ever Use | No | 124 | 15.4 | 285 | 35.39 | 396 | 49.21 | 805 |
| | Yes | 205 | 15.68 | 463 | 35.41 | 640 | 48.92 | 1309 |
| | Total | 329 | 15.57 | 748 | 35.4 | 1036 | 49.03 | 2114 |

*(Continued)*

**Table 2.** (Continued)

| Variable | | Mixed Feelings or Unhappy | % | Sort Happy | % | Very Happy | % | Total |
|---|---|---|---|---|---|---|---|---|
| Residence | Urban | 51 | 10.56 | 148 | 30.89 | 280 | 58.55 | 479 |
| | Rural | 279 | 17.04 | 601 | 36.71 | 757 | 46.25 | 1636 |
| | Total | 329 | 15.57 | 749 | 35.4 | 1037 | 49.03 | 2115 |
| Region | Tigray and Afar | 15 | 7.88 | 39 | 19.91 | 141 | 72.21 | 196 |
| | Amhara | 36 | 8.59 | 179 | 43.22 | 199 | 48.18 | 414 |
| | Oromiya | 185 | 20.67 | 267 | 29.72 | 445 | 49.6 | 897 |
| | SNNPR*/* | 84 | 16.02 | 238 | 45.27 | 204 | 38.71 | 527 |
| | Addis Ababa | 9 | 10.46 | 26 | 31.46 | 48 | 58.08 | 82 |
| | Total | 329 | 15.57 | 749 | 35.4 | 1037 | 49.03 | 2115 |
| Desired Birth Attendant | No One | 20 | 16.87 | 41 | 35.13 | 57 | 48 | 118 |
| | Health Professional | 193 | 13.73 | 475 | 33.83 | 737 | 52.44 | 1405 |
| | Family Member | 115 | 19.43 | 232 | 39.29 | 244 | 41.28 | 590 |
| | Total | 328 | 15.5 | 749 | 35.42 | 1037 | 49.08 | 2113 |
| Desired Delivery Place | Home | 137 | 18.33 | 290 | 38.78 | 321 | 42.89 | 748 |
| | Government and Private Health Facility | 192 | 14.06 | 459 | 33.54 | 716 | 52.39 | 1367 |
| | Total | 329 | 15.57 | 749 | 35.4 | 1037 | 49.03 | 2115 |

*/* = former Southern Nations Nationalities and Peoples Region which consists of the current South Ethiopia, Central Ethiopia, South West Ethiopia, and Sidama Regions, * = Wakefeta and traditional religion.

The likelihood of perceived paternal emotional fertility intention of being very happy was found to be (AOR; 95%CI: 5.06 (1.73, 14.85) and (AOR; 95%CI: 2.65 (1.67, 4.20) times higher among older women aged 40 to 49 years and among the pregnant women who wanted to have an additional child after giving birth of the index child respectively. On the contrary, those with higher birth order with 3 to 12 children had 86% (AOR: 95%CI; 0.14 (0.07, 0.27) and those who no more need another child had 46% 0.54 (AOR: 95%CI; 0.33, 0.90) lower likelihood of perceived paternal emotional fertility intention of being very happy (Table 3).

Similarly, this perceived likelihood on paternal emotional fertility intentions of being very happy was found to be 66% lower (AOR: 95%CI; 0.34 (0.17, 0.67) for those living as a partner. Regarding region of residence pregnant women had lower likelihood to perceive their husbands and/or partners as very happy when they learnt about the index pregnancy: for Addis Ababa it was 73% lower (AOR: 95%CI; 0.27 (0.14, 0.53) and 75% lower (AOR: 95%CI; 0.25 (0.15, 0.40) for residents of SNNPR compared with residents of Tigray and Afar regions.

On the other hand, the likelihood of perceived paternal emotional fertility intention of being a sort of happy was found to be (AOR: 95%CI; 1.93 (1.21, 3.10) times higher among those who wanted to have another child (Table 3). On the contrary, those with higher birth order and those who did not wanted to have an additional child had 57% (AOR: 95%CI; 0.43 (0.22, 0.85) and 42% (AOR: 95%CI; 0.58 (0.35, 0.97) lower likelihood of perceived paternal emotional fertility intention of being a sort of happy about the index pregnancy. This likelihood was 55% lower (AOR: 95%CI; 0.45 (0.28, 0.74) among residents of Oromiya region compared with residents of Tigraye and Afar Region. Moreover, women whose desired place of delivery was a heath facility had a 38% (AOR: 95%CI; 0.62 (0.43, 0.89) lower likelihood of perceiving that their husband felt a sort of happy when learnt about the index pregnancy (Table 3).

**Table 3. Multinomial regression modeling for of perceived paternal emotional fertility intentions about the index pregnancy among a panel of pregnant women and its correlates in Ethiopia, Nov 19 to Jan 20.**

| Independent Variables | Variables | Sort Happy (ARRR) | Very Happy (ARRR) |
|---|---|---|---|
| Women Age Category | 15–19 years | 1 | 1 |
| | 20–24 years | 0.84 (0.41,1.74) | 1.09 (0.54,2.19) |
| | 25–29 years | 1.21 (0.56,2.64) | 1.95 (0.92, 4.51) |
| | 30–34 years | 1.09 (0.47,2.55) | 2.38 (1.04, 5.44)* |
| | 35–39 years | 1.25 (0.52,3.06) | 2.36 (0.98, 5.67) |
| | 40–49 years | 1.12 (0.37,3.44) | 5.06 (1.73, 14.85)** |
| Educational Status | No formal education | 1 | 1 |
| | Primary Education | 0.75 (0.51,1.09) | 0.90 (0.62,1.31) |
| | Secondary+_Education | 0.65 (0.37, 1.12) | 1.26 (0.74.2.13) |
| *Wealth Index* | Poorest quintile | 1 | 1 |
| | Lower quintile | 0.86 (0.53, 1.41) | 1.05 (0.64,1.73) |
| | Middle quintile | 1.02 (0.62, 1.69) | 1.32 (0.79,2.19) |
| | Higher quintile | 1.47 (0.86, 2.52) | 1.48 (0.86,2.55) |
| | Highest quintile | 1.84 (0.87,3.89) | 1.41 (0.67,2.96) |
| Parity | No Child | 1 | 1 |
| | 1_2 Children | 0.49 (0.28, 0.87)* | 0.26 (0.15, 0.45)*** |
| | 3_12 Children | 0.43 (0.22, 0.85)* | 0.14 (0.07, 0.27)*** |
| Fertility Desire | Undecided | 1 | 1 |
| | Wanted to have another child | 1.93 (1.21,3.10)** | 2.65 (1.67,4.20)*** |
| | No more to have another child | 0.58 (0.35, 0.97)* | 0.54 (0.33, 0.90)* |
| Marital Status | Married | 1 | 1 |
| | Living as a partner | 0.56 (0.28,1.11) | 0.34 (0.17,0.67)** |
| Contraceptive Ever Use | No | 1 | 1 |
| | Yes | 0.87 (0.61, 1.25) | 0.97 (0.68,1.73) |
| *Residence* | Urban | 1 | 1 |
| | Rural | 1.37 (0.77, 2.46) | 0.98 (0.56,1.39) |
| Region | Tigray & Afar | 1 | 1 |
| | Amhara | 1.71 (0.95, 3.06) | 0.45 (0.25,0.80)** |
| | Oromiya | 0.45 (0.28, 0.74)** | 0.21 (0.13,.33)*** |
| | SNNP | 1.01 (0.62,1.64) | 0.25 (0.15,0.40)*** |
| | Addis | 0.92 (0.46,1.88) | 0.27 (0.14,0.53)*** |
| Desired Delivery Place | Home | 1 | 1 |
| | Health Facility | 0.62 (0.43,0.89)** | 0.80 (0.53,1.16) |

*** $p < .001$, ** $p < 0.01$, * $p < 0.05$.

## Discussion

There is no strategic and legal policy framework for males' involvement during pregnancy and childbirth care including providing emotional care and support in Ethiopia. Moreover, males' dominance in household decision making in general and on reproductive health and fertility desire decisions in particular is pronounced. Hence, determining the level of perceived paternal emotional fertility intentions about the index pregnancy and its correlates among a cohort of pregnant women is a very critical step to improve maternal and new born health outcomes. Such a dominance is culturally accepted and socially constructed and reflected in the lack of emotional care and support during pregnancy and childbirth by husband and/or partner.

Being short of halfway towards the SDG period, measuring perceived partner emotional fertility intention among pregnant women as a measure of couples reproductive health right and decision making on their reproductive life and/or health is hoped to generate and provide an actionable evidence for the ministry and relevant actors to improve women decision making on their desired fertility in particular and their reproductive health right in general. It also provided relevant information to enhance emotional wellbeing of pregnant women during pregnancy and improve emotional care and support by husbands during pregnancy and childbirth.

The finding that half of the pregnant women perceived that their husband and/or partner felt very happy or one third felt a sort of happy was in agreement with the finding on fathers' engagement in pregnancy and childbirth: evidence from a national survey from England which found that over 80% of fathers were 'pleased´ or 'overjoyed' in response to their partner's pregnancy; and over half were present for the pregnancy test; for one or more antenatal checks, and almost all were presented for ultrasound examinations and for labor [41]. On the contrary, a study on first-time fathers' study: psychological distress in expectant fathers during pregnancy [37] reported that fathers who had insufficient information about pregnancy and childbirth were also at risk of being distressed, suggesting that more attention needed to be paid to provide information to men about their wives pregnancy, childbirth and issues relating to caring for a newborn infant. This finding implied that activities need to framed and efforts need to be made to improve family health and couples communication as well as creating conducive environment for men to be involved in in their wives´ pregnancies by providing emotional care and support and articulating and endorsing strategy for their involvement as well is warranted [14]. This contributed for the cumulative effort in creating a happy family and maintaining developmentally sound fertility rates in Ethiopia. Compared with a study among US men which reported that 63% of births were intended by the father [42] this study´s finding (49.03%; 46.48%, 51.6%) was found to be lower. This same study reported 62% of births received a 10 on the happiness scale and another 18% received a value of 7–9, indicating the father had been highly happy upon learning about their wives pregnancies. This discrepancy might be due to measurement approach difference: the cited study measured how father felt about births they had during the last 5 years while our study measured pregnant women perceived paternal emotional fertility intention about the index pregnancy. Another study reported that 646 (81%) men stated that the pregnancy had been very or fairly planned [32] which is higher than this study finding which might be explained by the variation in how the outcome variables were measured.

The higher paternal perceived emotional fertility intention likelihood of being happy among elders might be likely to emanate from their previous pregnancy and child bearing experience leading them to be well prepared for their index pregnancies [43] and as well as might be highly related with good couples interaction and communication [19,44]. Similarly, the finding that pregnant women who intended to have another child had an increased likelihood to perceive that their husband and/or partner felt happy when learning their index pregnancy might be related with the families' aspiration to achieve the desired family size and can be seen as exercising their reproductive health rights and reproductive autonomy [45].

On the contrary, the finding that lower likelihood of pregnant women to perceive that their husband and/or partner felt happy when they learned about the index pregnancy among those with higher birth order and those who do not want any more child might be related to women prior pregnancy experience [43] and is also likely related with males´ decision making on the number of children; as wells women became pregnant because of their husband and/or partner influence as husbands/partners need more children [16,46,47]. In addition, the finding that higher birth order lowers pregnant women likelihood of being

happy with paternal emotional fertility intentions was in line with findings from studies on women fertility desire [48,49].

The regional variation in perceived paternal emotional fertility intentions of being happy when learning the index pregnancy; was not in line with a study on actual fertility desire. The other major finding of this study was region of residence lowered the likelihood of perceived paternal emotional fertility intention of being happy while a study on women´s fertility [50] showed an increased association between the region of residence with a higher desire for more children [50]. The discrepancy might be due to the fact that the difference in study population and how the outcome variable was measured. By implication their perceived paternal emotional fertility intention might be the same which is coupled with cultural and social acceptability of males' dominance [14,33–35,51].

Unlike this study, a study [42] reported that marital status and age of fathers affected contributed for the variation in perceived paternal emotional fertility intention about live births that the father had 5 years preceding the survey. Another study [32] reported that education improves men´s fertility happiness through live style change and better education.

The study´s implication is increasing men fertility knowledge and their emotional readiness as supported by a finding from a study that men with university education had better fertility knowledge than men without a university education alongside lifestyle adjustment before pregnancy to improve health and fertility [32]. In addition, more educated husbands tend to understand their wives, communicate smoothly, support and accompany them to the health facility while seeking person centered care during pregnancy and childbirth care [52]. The other implication of the study finding was that it provided its share for developing and implementing males' involvement strategy during pregnancy, childbirth and related reproductive health service use including the emotional care and support.

This study is not spared of limitations. To start with though reliance on self-reported data by asking the women to learn about how their husbands felt about the index pregnancies to measure paternal emotional fertility intention leads to potential biases such as social desirability bias as not talking negatively about once husband and/or partner has been a customer practice in our context, such findings offer a meaningful insights which is important and relevant to outlining care for pregnant women and their partners. However, further research is needed to validate and build upon these initial results including studies that come up with alternative and direct measures of paternal emotional fertility intention. In addition, because the PMA 2019/20 baseline survey didn't collect information on variables such as husband desired number of children, husband employment and women employment were not measured in this study. Further research aimed at exploring perceived paternal fertility emotions across different cultural or socio-economic groups are need.

## Conclusion

Only half of the pregnant women perceived their husband felt happy about their index pregnancy calls up on improving couples communication, and discussion on the spacing and timing of pregnancy as well as to work on improving birth preparedness and complication readiness. The finding also calls up on creating awareness on preconception care packages in and around pregnancy and to avail the preconception care service which are critical cares before pregnancy, between pregnancies and as a life course approach. These activities and efforts need to be age and region specific. The finding of the study implied that the ministry and relevant partners need to work strategically on male involvement in pregnancy, child birth and fertility desire along with emotional care and support. These parties need to design and implement age sensitive and region specific interventions that target women with future fertility intention, lower birth orders and those who have not been legally married

and targeted in enhancing facility and skilled delivery care. The implication of this study is installing preconception care particularly the interpregnancy care package along with improving counseling and provision of postpartum contraceptives. Such interventions are hoped to greatly improve perceived paternal emotional fertility intentions of being happy when learning the index pregnancy. The other implication of this study is increasing men fertility knowledge and their emotional readiness along with lifestyle adjustment before pregnancy to improve health and emotional fertility intentions. Lastly, birth preparedness and complication readiness would help husbands and/or partners thereby improving their actual fertility intention and would help their wives to better perceive them in their involvement during childbirth as of the planning stage. Further research is needed to explore perceived paternal fertility emotions across different cultural or socio-economic groups. Additional studies are needed to come up with a direct measurement of paternal emotional fertility intention including validating this direct measurement tool to be developed.

The critical implication of this study was providing part of the evidence for WHO´s recommendation on husband involvement in maternal and newborn care service uptake thereby improving maternal and newborn outcomes. It also provide evidences to design policy and strategy on husbands' involvement during pregnancy, childbirth care; along with care and support for pregnant and postpartum women. It's also provided evidence for couple's fertility control and the role of husband on couples inter pregnancy contraception use. Furthermore health care providers need to take the opportunity of antenatal and childbirth care visits to arrange discussion with couples on the importance of the husband emotional support during pregnancy and childbirth.

## Acknowledgement

We acknowledge the PMA Ethiopia project data-collecting team members and research participants.

## Author contributions

**Conceptualization:** Solomon Abrha Damtew.

**Data curation:** Solomon Abrha Damtew, Tariku Dejene Demissie.

**Formal analysis:** Solomon Abrha Damtew.

**Funding acquisition:** Assefa Seme, Solomon Shiferaw.

**Investigation:** Solomon Abrha Damtew, Assefa Seme, Solomon Shiferaw.

**Methodology:** Solomon Abrha Damtew, Assefa Seme, Solomon Shiferaw.

**Project administration:** Solomon Abrha Damtew, Mahari Yihdego Gidey, Fitsum Tariku Fantaye, Niguse Tadele Atnafu, Hailay Gebremichael Gebrekidan, Tariku Tesfaye Bekuma, Assefa Seme, Solomon Shiferaw.

**Resources:** Assefa Seme, Solomon Shiferaw.

**Software:** Solomon Abrha Damtew, Tariku Dejene Demissie.

**Supervision:** Solomon Abrha Damtew, Mahari Yihdego Gidey, Niguse Tadele Atnafu, Kelemua Mengesha Sene, Tariku Dejene Demissie, Assefa Seme, Solomon Shiferaw.

**Validation:** Solomon Abrha Damtew, Mahari Yihdego Gidey, Niguse Tadele Atnafu, Tariku Dejene Demissie, Assefa Seme, Solomon Shiferaw.

**Visualization:** Solomon Abrha Damtew, Tariku Dejene Demissie.

**Writing – original draft:** Solomon Abrha Damtew.

**Writing – review & editing:** Solomon Abrha Damtew, Mahari Yihdego Gidey, Fitsum Tariku Fantaye, Niguse Tadele Atnafu, Kelemua Mengesha Sene, Bezawork Ayele Kassa, Hailay Gebremichael Gebrekidan, Tariku Tesfaye Bekuma, Seifu Yennedu Berhe, Gelane Duguma Edosa, Temesgen Bati Gelgelu, Dereje Haile, Wakgari Binu Daga, Tesfamichael Awoke Sisay, Ayanaw Amogne, Tariku Dejene Demissie, Assefa Seme, Solomon Shiferaw.

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
