## [Decision Letter · Decision Letter 0]

4 Sep 2024

PONE-D-24-25872Perceived Paternal Fertility Emotion and Its correlates in Ethiopia among a Cohort of Pregnant Women: Community based Longitudinal SurveyPLOS ONE

Dear Dr. Damtew,

Thank you for submitting your manuscript to PLOS ONE. After careful consideration, we feel that it has merit but does not fully meet PLOS ONE’s publication criteria as it currently stands. Therefore, we invite you to submit a revised version of the manuscript that addresses the points raised during the review process.

We look forward to receiving your revised manuscript.

Kind regards,

Yibeltal Alemu Bekele, MpH

Academic Editor

PLOS ONE

Journal Requirements:

4. Please include a separate caption for figure 1. 

5. We notice that your supplementary figures & tables are included in the manuscript file. Please remove them and upload them with the file type 'Supporting Information'. Please ensure that each Supporting Information file has a legend listed in the manuscript after the references list.

Reviewers' comments:

Reviewer's Responses to Questions

**Comments to the Author**

1. Is the manuscript technically sound, and do the data support the conclusions?

Reviewer #1: Yes

Reviewer #2: Partly

2. Has the statistical analysis been performed appropriately and rigorously? 

Reviewer #1: Yes

Reviewer #2: Yes

3. Have the authors made all data underlying the findings in their manuscript fully available?

Reviewer #1: Yes

Reviewer #2: Yes

4. Is the manuscript presented in an intelligible fashion and written in standard English?

Reviewer #1: Yes

Reviewer #2: No

5. Review Comments to the Author

Reviewer #1: Title:

- Please include that this is a secondary data analysis and specify the year of the study.

Abstract:

- Results Section: There is an inconsistency regarding higher birth order. The authors report that having 3 to 12 children (AOR=0.14) is associated with higher birth order, yet at the end of the results section, they state that higher birth order was a significant predictor of being happy about the index pregnancy. Please clarify this inconsistency.

Background:

- The introductory paragraph could benefit from restructuring. I recommend following a funnel approach to better organize the background information.

Language and Style:

- The manuscript would benefit from improvements in English language grammar and overall writing style.

Methodology:

- Sample Population: Provide additional details about the PMA (Performance Monitoring for Action) including the sampling process and the type of data collected.

- Study Design: The authors describe this study as a cohort study. However, it is unclear if the pregnant women were followed over time to assess any changes in their partner's emotions towards the pregnancy. Please clarify this aspect.

- Data Collection: The rationale for accessing data at 6 weeks postpartum is unclear. Were participants questioned about their pregnancies post-delivery? The absence of reporting on this second data point suggests that this might be more accurately described as a cross-sectional study, where a group of women was surveyed about their pregnancies at a single point in time.

Sample Size:

- Include a flowchart diagram illustrating the sample selection process and how the final sample size was determined.

Independent Variables:

- Please enumerate the individual-level and enumeration area-level variables used in this study.

Discussion:

- The second paragraph is unnecessary and poorly written.

- Overall, the discussion section is underdeveloped and requires significant improvement.

Reviewer #2: Dear authors,

congratulations on your study. The subject is important and relevant to outlining care for pregnant women and their partners. Some suggestions listed below could strengthen your manuscript.

In the introduction:

1. I was unsure if ‘Intention to conceive’ is the same as ‘Intention to fertility’? You talk about these expressions and they seem to be synonyms, but they can be totally different when you investigate the meaning of the words. If you're referring to different processes, I'd gently suggest making that clearer in the text. A brief adjustment to the paragraph is sufficient.

On the method:

2. I'm understanding that the aim of the study was to determine the level of perceived paternal fertility emotion about the index pregnancy among a cohort of pregnant women and to identify their correlates for such variation, right? But when I read the method, I realise that the data was collected from women. I wonder if I've understood the objective correctly, because how can we measure the perception of fathers through mothers? Shouldn't this information be collected from men?

3. What was the design of the study presented in this article? Was it a cross-sectional study with qualitative data? or quantitative? or mixed method? I strongly suggest signalling this right at the beginning of the method passage.

4. Although the data has already been published, providing a brief contextualisation of the study makes it easier to understand. I'm not saying that you have to write everything down in exhaustive detail. Just contextualise it simply and objectively: this is a cross-sectional study of the baseline of a prospective cohort. The data was collected from pregnant women in X health centres in Ethiopia....

5. Where you reference the details of the Ethiopia PMA, I suggest that you leave only the citation of the study and move the reference to the references section. For example: ‘A detailed description of the Ethiopia PMA protocol has been published previously (citation)’ or ‘Information regarding the selection of the sample and the units is described in detail in the Ethiopia PMA protocol (citation)’.

6. I kindly suggest that you objectively describe which sample was used in this article. It doesn't matter if you recruited people with a different profile at other points in the study. What matters is that you make it clear what the inclusion and sampling criteria were for the section of the study you are presenting here. For example: puerperal women were recruited throughout the study, but only pregnant women were recruited at baseline. So you can go straight to the information that the sample consists of pregnant women.

Results:

The findings are very good. The quality of the data is excellent, given the size of the sample and the categories of responses. It's positively surprising!

7. It would be very comfortable for the reader if you could include the number of participants who were included in the analysis in the second line of the results. Even if it is already well described in the method, just for comfort in reading and interpreting. ‘2115 women took part in the study. The proportion of parental emotion....’ is enough.

8. In the passage: Correlates of Perceived Paternal Fertility Emotion about the Index Pregnancy among a panel of pregnant women and its correlates in Ethiopia, Evidence from Cohort One Baseline Data, 2019 to 20. The first sentence “This study investigated factors affecting perceived paternal fertility emotion about the index pregnancy among a cohort of pregnant when their husband and/or partner learned their wife´s index pregnancy” fits better at the end of the introduction or in an objective passage. You don’t have to rewrite it in the results because it should be clear in the method. So I truly suggest you exclude this sentence in this paragraph, but consider bringing it to the end of the introduction or at the beginning of the method.

Discussion:

9. In the second paragraph of the discussion you cited “the authors” two times. It is not clear if you are talking about the authors of the dataset or if you are talking about you (authors of the manuscript) in a third voice. If you are talking about the dataset’s authors, I strongly suggest you provide the citation in this paragraph. If you are not, I suggest making it clear that you are discussing your findings. This passage is a little confusing.

10. What limitations did the study have? Think about reiterating that paternal feelings were assessed through women's reports and are therefore subject to bias. How can someone else tell what I'm feeling?

Conclusion:

11. I totally agree with your point of view, but you should respond to your study’s objective here. The other statements about what to do with these findings must be discussed in the discussion passage.

12. the English is a little hard to follow. An English language review would be nice.

6. PLOS authors have the option to publish the peer review history of their article (what does this mean? ). If published, this will include your full peer review and any attached files.

**Do you want your identity to be public for this peer review?** For information about this choice, including consent withdrawal, please see our Privacy Policy .

Reviewer #1: **Yes: ** Nebyu Amaha

Reviewer #2: No

---

## [Author Response · Author response to Decision Letter 1]

19 Sep 2024

Date: 19 Sep 2024

To Plos One

Subject: Point by Point Response

Manuscript Id: PONE-D-24-25872

Title Perceived Paternal Fertility Emotion and Its correlates in Ethiopia among a Cohort of Pregnant Women: Community based Longitudinal Survey

Journal: PLOS ONE

The authors very much appreciate for the constructive and valuable comments provided by the reviewers which guided the authors to improve the quality of the revised manuscript. We appreciated all comments and used as an input to improve the quality of the manuscript without such feedback the revised manuscript would not take its current form. We also extend our sincere acknowledgment to handling the editor for giving us the opportunity to further improve the quality of the manuscript so that it will suitable for publication.

Response to Reviewer one valuable Concerns:

Reviewers' comments:

Response to Reviewer 1:

Reviewer #1:

General Response: We very much appreciate the reviewer’s constructive feedback and suggestion. We have attempted to provide responses for the specific queries.

Comment 1: Title: - Please include that this is a secondary data analysis and specify the year of the study.

Response 1: We appreciate the comment and comment addressed in the revised version.

Revised Title Perceived Paternal Fertility Emotion and Its correlates in Ethiopia among a Cohort of Pregnant Women: Community based Longitudinal Survey; A secondary data analysis of 2019/20 Baseline Survey

Abstract:

Comment 2: - Results Section: There is an inconsistency regarding higher birth order. The authors report that having 3 to 12 children (AOR=0.14) is associated with higher birth order, yet at the end of the results section, they state that higher birth order was a significant predictor of being happy about the index pregnancy. Please clarify this inconsistency.

Response 2: We have double checked the twos sections and found no discrepancy for the very happy categories of the outcome since we have two categories for the outcome to be compared internally with the base category. In both sections higher birth order lower perceived paternal fertility emotion between the what is stated in the result part of the abstract and the main result part: page 11 paragraph one: summary and paragraph four narration about the specific variable. This applies for the very happy category.

Note that; we have two upper level categories; the very happy and a sort of happy category for ARRR is reported as compared with the base category of the outcome. 3 to 12 children (AOR=0.14): this is for the very happy category while the later stated as significant predictors and the effect measure for the sort of happy category to which the specific ARRR is plugged in with the abstract in the revised version.

Background:

Comment 3: - The introductory paragraph could benefit from restructuring. I recommend following a funnel approach to better organize the background information.

Response 3: Comment accepted and efforts were made to rearrange the background information so that it will at least resembles the funnel approach suggested by the reviewer.

Language and Style:

Comment 4: - The manuscript would benefit from improvements in English language grammar and overall writing style.

Response 4: We have attempted to get supported from English Language Professional Colleague.

Methodology:

Comment 5: - Sample Population: Provide additional details about the PMA (Performance Monitoring for Action) including the sampling process and the type of data collected.

Response 5: PMA employed a two cluster stage sampling; in the first stage enumeration areas (EA) were selected. In each selected EA fresh listing and census was conducted to screen and enroll pregnant women and less than 6 weeks postpartum women by the time of the baseline survey those this study was restricted to only 2,115 pregnant women from 217 enumerations areas who were married and/or living together as a partner by the time of the base line survey and who completed the baseline female questionnaire. Following enrollment the Female baseline questionnaire was administered. In the female baseline questionnaire women were asked about their antenatal care sought thus far in their idex pregnancy, partner support and perceived community encouragement on the use the three main domains of the maternal and newborn care components, their reproductive and sexual history, their birth preparedness and complication readiness, about their agreement and disagreement on girls and women empowerment towards contraceptive use; women sexual and reproductive issues, contraceptive use history and there fertility intention. Women were asked how they themselves did felt when learned the index pregnancy. Most importantly to this study, they were also asked how their husband felt when their husbands and/or partners learned their index pregnancy, which regarded as women perceived paternal fertility emotion towards the index pregnancy which is the main outcome variable in this study.

Comment 6: - Study Design: The authors describe this study as a cohort study. However, it is unclear if the pregnant women were followed over time to assess any changes in their partner's emotions towards the pregnancy. Please clarify this aspect.

Response 6: We appreciate the comment. Sure the actual study continued to one postpartum with 6 weeks, 6 months and one year postpartum follow up after the baseline survey. This study mainly rely on the question how their husband felt when leaned the index pregnancy, neither the husbands were asked. This why we use the term perceived. Hence, we include the following limitations:

Though reliance on self-reported data by asking the women to learn about how their husband felt about the index pregnancy to measure on paternal emotional fertility intention which lead to potential biases particularity; such findings offer meaningful insights that advance the field of study. However, further research is needed to validate and build upon these initial results.

For your information women were asked what did the feel if they became pregnant in the follow up interviews.

Comment 7: - Data Collection: The rationale for accessing data at 6 weeks postpartum is unclear. Were participants questioned about their pregnancies post-delivery? The absence of reporting on this second data point suggests that this might be more accurately described as a cross-sectional study, where a group of women was surveyed about their pregnancies at a single point in time.

Response 7: Thank you very much for the question posed, accessing six weeks postpartum date set was mention in the Ethical consideration section, page 8. For your information, this data set were not used in this particular analysis and instead only the baseline cross-sectional data set from the prospective study were analyzed!

Sample Size:

Comment 8: - Include a flowchart diagram illustrating the sample selection process and how the final sample size was determined.

Response 8: Two thousand two hundred thirty nine (2,239) women provided response for the When your partner found out you were pregnant, how did he feel, 124 were exclude from the analysis due to 3 women provided no response, 46 reported do not know, 57 reported that they have not told to their partner and 18 of them reported they have no partner. Hence, the final sample size was 2115.

Independent Variables:

Comment 9: - Please enumerate the individual-level and enumeration area-level variables used in this study.

Response 9: Comment accepted and the following revision was made in the revised manuscript version.

Sociodemographic: age of women, educational status, wealth index, marital status and religion of the women. RH characteristics: birth order, marriage type, marriage history, fertility intention, health service use related such as contraceptive ever use history and planned birth attendant and planed place of delivery for the index child. The integral enumerating area level variables were region and place of residence. Note that derived enumeration area level variables were EA level wealth and proportion of women with secondary education and above. Here below is the list of independent variables included in this study along with the final recode categories,

Variable name and Categories

Age category 15-19 years

20-24 years

25-29 years

30-34 years

35-39 years

40-49 years

Educational Status No Formal Education

Primary Education

Secondary Plus

Religion Other

Orthodox

Protestant

Muslim

Wealth Index Lowest quintile

Lower quintile

Middle quintile

Higher quintile

Highest quintile

Parity No Child

1_2 Children

3_12 Children

Marriage History Only_ Once

More than_ Once

Marriage Type Monogamy

Polygamy

Fertility Intension Undecided/DKN

Have a/another child

No more or prefer no child

Marital Status Married

Living With A partner

Widowed/Separated

Contraceptive Ever Use No

Yes

Residence Urban

Rural

Region Tigray and Afar

Amhara

Oromiya

SNNPR

Addis Ababa

Desired Birth Attendant No One/ND

HP

Family Member

Desired Delivery Place Home/ND

Government Private HF

Discussion:

Comment 10: - The second paragraph is unnecessary and poorly written.

Response 10: Comment accepted and efforts were made to amend the second paragraph.

Comment 11: - Overall, the discussion section is underdeveloped and requires significant improvement.

Response 11: Comment accepted and efforts were made to enrich the discussion part, which is all related with the dearth of literature directly fit the study measurement approach. Efforts were made to enrich the discussion part in the revised version. Additional empirical evidence was sought and cited.

Response to Reviewer 2:

Reviewer #2: Dear authors,

Congratulations on your study. The subject is important and relevant to outlining care for pregnant women and their partners. Some suggestions listed below could strengthen your manuscript.

General Response: We appreciate the encouragement and for the reviewer´s kind words. We have attempted to provide responses for the specific queries.

In the introduction:

Comment 1: 1. I was unsure if ‘Intention to conceive’ is the same as ‘Intention to fertility’? You talk about these expressions and they seem to be synonyms, but they can be totally different when you investigate the meaning of the words. If you're referring to different processes, I'd gently suggest making that clearer in the text. A brief adjustment to the paragraph is sufficient.

Response 1: We considered the reviewers comment and preferred the term fertility intention over intention to conceive which is corrected in the revised version.

On the method:

Comment 2: 2. I'm understanding that the aim of the study was to determine the level of perceived paternal fertility emotion about the index pregnancy among a cohort of pregnant women and to identify their correlates for such variation, right? But when I read the method, I realize that the data was collected from women. I wonder if I've understood the objective correctly, because how can we measure the perception of fathers through mothers? Shouldn't this information be collected from men?

Response 2: We acknowledge this concern as it’s related with outcome variable measurement. As you rightly mentioned, It could have been better had the information was collected from men. Since Such data were scarce and we relied on secondary data we used the term perceived paternal fertility intention as his feeling towards the index pregnancy were measured by asking his wife which was used as a proxy measure. This is acknowledge as a limitation of this study.

This study is not spared of limitations. To start with though reliance on self-reported data by asking the women to learn about how their husband felt about the index pregnancy to measure on paternal emotional fertility intention which lead to potential biases; such findings offer meaningful insights which is important and relevant to outlining care for pregnant women and their partners. However, further research is needed to validate and build upon these initial results. In addition, because of PMA 2020 didn’t collect information on variables such as husband desired number of children, husband employment and women employment were not measured in this study.

Comment 3: 3. What was the design of the study presented in this article? Was it a cross-sectional study with qualitative data? or quantitative? or mixed method? I strongly suggest signaling this right at the beginning of the method passage.

Response 3: Comment accepted and the following introduction signaling the specific data source and design used in this study plugged in: This study used the baseline cross-sectional data from prospective cohort study with 6 weeks, 6 months and one postpartum follow up interviews apart from the baseline cross sectional data used for this study. The data was collected from pregnant women in Ethiopia from six regions: namely: Addis Ababa, Afar, Amhara, Oromia, SNNPR and Tigray.

Comment 4: 4. Although the data has already been published, providing a brief contextualization of the study makes it easier to understand. I'm not saying that you have to write everything down in exhaustive detail. Just contextualize it simply and objectively: this is a cross-sectional study of the baseline of a prospective cohort. The data was collected from pregnant women in X health centres in Ethiopia....

Response 4: We appreciate the comment and the following introduction at the beginning of the methods part was used: this study used the community based baseline cross-sectional data from prospective cohort study with 6 weeks, 6 months and one postpartum follow ups interviews. The data was collected from pregnant women in Ethiopia from six regions: namely: Addis Ababa, Afar, Amhara, Oromia, SNNPR and Tigray.

Comment 5: 5. Where you reference the details of the Ethiopia PMA, I suggest that you leave only the citation of the study and move the reference to the references section. For example: ‘A detailed description of the Ethiopia PMA protocol has been published previously (citation)’ or ‘Information regarding the selection of the sample and the units is described in detail in the Ethiopia PMA protocol (citation)’.

Response 5: Comment accepted and corrected in the revised version of the manuscript.

Comment 6: 6. I kindly suggest that you objectively describe which sample was used in this article. It doesn't matter if you recruited people with a different profile at other points in the study. What matters is that you make it clear what the inclusion and sampling criteria were for the section of the study you are presenting here. For example: puerperal women were recruited throughout the study, but only pregnant women were recruited at baseline. So you can go straight to the information that the sample consists of pregnant women.

Response 6: we acknowledge your concerns and address this very important concern by including expiation in the methods part within the revised version The study started by recruiting and enrolling pregnant women and puerperal women less than six weeks which was followed admistrising the female baseline questionnaire. Then these panel of women were interviewed at 6 weeks, 6 months and one year postpartum as follow up interviews. However, this study further analyzed and present data from the baseline survey

Results:

The findings are very good. The quality of the data is excellent, given the size of the sample and the categories of responses. It's positively surprising!

Comment 7: 7. It would be very comfortable for the reader if you could include the number of participants who were included in the analysis in the second line of the results. Even if it is already well described in the method, just for comfort in reading and interpreting. ‘2115 women took part in the study. The proportion of parental emotion....’ is enough.

Response 7: Comment accepted and revision made in the revised version. Responses on how pregnant women perceived paternal fertility intention (what their husbands and/or partners did they felt when learned the index pregnancy) by the time they were pregnant from a total of 2,115 pregnant

---

## [Decision Letter · Decision Letter 1]

2 Dec 2024

PONE-D-24-25872R1Perceived Paternal Fertility Emotion and Its correlates in Ethiopia among a Cohort of Pregnant Women: Community based Longitudinal Survey; A Secondary Data Analysis of 2019/20 Baseline SurveyPLOS ONE

Dear Dr. Damtew,

Thank you for submitting your manuscript to PLOS ONE. After careful consideration, we feel that it has merit but does not fully meet PLOS ONE’s publication criteria as it currently stands. Therefore, we invite you to submit a revised version of the manuscript that addresses the points raised during the review process.

We look forward to receiving your revised manuscript.

Kind regards,

Yibeltal Alemu Bekele, MpH

Academic Editor

PLOS ONE

Reviewers' comments:

Reviewer's Responses to Questions

**Comments to the Author**

1. If the authors have adequately addressed your comments raised in a previous round of review and you feel that this manuscript is now acceptable for publication, you may indicate that here to bypass the “Comments to the Author” section, enter your conflict of interest statement in the “Confidential to Editor” section, and submit your "Accept" recommendation.

Reviewer #1: All comments have been addressed

2. Is the manuscript technically sound, and do the data support the conclusions?

Reviewer #1: Partly

3. Has the statistical analysis been performed appropriately and rigorously? 

Reviewer #1: Yes

4. Have the authors made all data underlying the findings in their manuscript fully available?

Reviewer #1: No

5. Is the manuscript presented in an intelligible fashion and written in standard English?

Reviewer #1: Yes

6. Review Comments to the Author

Reviewer #1: Abstract

1. The term "fertility emotion" should be defined briefly to ensure clarity for readers unfamiliar with the concept.

2. Clarify how the findings contribute to existing knowledge or practice.

Introduction

1. Clearly articulate the specific research gap this study addresses. While it is implied that limited research exists in this area, explicit evidence (e.g., citing prior studies or lack thereof) would strengthen the justification.

2. The research objectives should be stated more explicitly. For example:

"This study aims to explore the perceived fertility emotions of male partners and examine their correlates among pregnant women in Ethiopia."

3. Clearly define how "Paternal Fertility Emotion" concept was operationalized. Were validated tools used, or was a new tool developed? If a new tool was developed, describe the validation process.

Results

1. Ensure consistency in terminology. For instance, if terms like "positive fertility emotion" or "negative fertility emotion" are used, define them clearly and use them consistently throughout the manuscript.

Discussion

1. Clearly explain how this study addresses the identified gap in knowledge.

2. Discuss the implications of these findings for maternal and paternal health policies or interventions in Ethiopia not a copy-paste of an implication from another study!! “The implication of this study is increasing men fertility knowledge and their emotional readiness as supported by a finding from a study Men with university education had better fertility knowledge than men without university education along with lifestyle adjustment before pregnancy to improve health and fertility (32)”

3. The limitations section should be expanded to address potential biases (e.g., social desirability bias in responses) and generalizability concerns.

4. Suggest specific areas for future research, such as exploring fertility emotions across different cultural or socio-economic groups.

7. PLOS authors have the option to publish the peer review history of their article (what does this mean? ). If published, this will include your full peer review and any attached files.

**Do you want your identity to be public for this peer review?** For information about this choice, including consent withdrawal, please see our Privacy Policy .

Reviewer #1: **Yes: ** Nebyu Daniel Amaha

---

## [Author Response · Author response to Decision Letter 2]

5 Dec 2024

Date: 04 Dec 2024

To Plos One

Subject: Point by Point Response

Manuscript Id: PONE-D-24-25872

Title Perceived Paternal Emotional Fertility Intention and Its correlates in Ethiopia among a Cohort of Pregnant Women: Community based Longitudinal Survey; A Secondary Data Analysis of 2019/20 Baseline Survey

Journal: PLOS ONE

The authors very much appreciate for the constructive and valuable comments provided by the reviewers which guided the authors to improve the quality of the revised manuscript. We appreciated all comments and used as an input to improve the quality of the manuscript without such feedback the revised manuscript would not take its current form. We also extend our sincere acknowledgment to handling the editor for giving us the opportunity to further improve the quality of the manuscript so that it will suitable for publication.

Response to Reviewer one valuable Concerns:

Reviewers' comments:

Reviewer #1: Abstract

Comment 1: 1. The term "fertility emotion" should be defined briefly to ensure clarity for readers unfamiliar with the concept.

Response 1: The authors appreciate the comment and amendment made in the revised version. Revisions made accordingly.

Comment 2: 2. Clarify how the findings contribute to existing knowledge or practice.

Response 2: The authors appreciate the comment and amendment made in the revised version in the implication of a study. The existing justification has the significance of the study narrated within it.

Introduction

Comment 3: 1. Clearly articulate the specific research gap this study addresses. While it is implied that limited research exists in this area, explicit evidence (e.g., citing prior studies or lack thereof) would strengthen the justification.

Response 3: The authors appreciate the comment and amendment made in the revised version. As narrated in the background justification and picked up by the reviewer the later the explanation worked.

Comment 4: 2. The research objectives should be stated more explicitly. For example:

"This study aims to explore the perceived fertility emotions of male partners and examine their correlates among pregnant women in Ethiopia."

Response 4: The authors appreciate the comment and amendment made in the revised version the stated justification were made.

Comment 5: 3. Clearly define how "Paternal Fertility Emotion" concept was operationalized. Were validated tools used, or was a new tool developed? If a new tool was developed, describe the validation process.

Response 5: The authors appreciate the comment and amendment made in the revised version. The authors would appreciated this study can service a preliminary finding in the area which could be capitalized by further studies. The following recommendation was added to address the aforementioned concern: Further research aimed at exploring perceived paternal fertility emotions across different cultural or socio-economic groups. Additional studies are needed which come up with a direct measurement of paternal emotional fertility intention including validating this direct measurement tool to be developed.

Results

Comment 6: 1. Ensure consistency in terminology. For instance, if terms like "positive fertility emotion" or "negative fertility emotion" are used, define them clearly and use them consistently throughout the manuscript.

Response 6: In the result section particularly the factors related section to mention show the direction of the statistical association, meaning those factors the increased the odds of perceived paternal emotional fertility intensions and those factors which were found to lower the it.

Discussion

Comment 7: 1. Clearly explain how this study addresses the identified gap in knowledge.

Response 7: We appreciate the comment and the significance of the study is presented in the background part. Additional relevance has been added as implication of the study.

Comment 8: 2. Discuss the implications of these findings for maternal and paternal health policies or interventions in Ethiopia not a copy-paste of an implication from another study!! “The implication of this study is increasing men fertility knowledge and their emotional readiness as supported by a finding from a study Men with university education had better fertility knowledge than men without university education along with lifestyle adjustment before pregnancy to improve health and fertility (32)”

Response 8: The authors appreciate the comment and amendment made in the revised version.

The following additional argument has been added….In addition, More educated husbands tend to understand their wives communicate smoothly and help them while seeking person centered care during pregnancy and childbirth care (52). The other implication of the study finding was that it provided its share for developing and implementing males’ involvement strategy in during pregnancy, childbirth and related reproductive health service use.

Comment 9: 3. The limitations section should be expanded to address potential biases (e.g., social desirability bias in responses) and generalizability concerns.

Response 9: The authors appreciate the comment and amendment made in the revised version. The following has been expanded in the limitation part as per the comment from the reviewer.

… Potential biases such as social desirability bias as talking negative about once husband and/or partner has been a customer practice in our context. ….

However, further research is needed to validate and build upon these initial results including studies which come with alternative and direct measures of paternal fertility emotional intention. In addition, because of PMA 2019/20 baseline survey didn’t collect information on variables such as husband desired number of children, husband employment and women employment were not measured in this study. Further research aimed at exploring perceived paternal fertility emotions across different cultural or socio-economic groups.

Comment 10: 4. Suggest specific areas for future research, such as exploring fertility emotions across different cultural or socio-economic groups.

Response 10: The authors appreciate the comment and amendment made in the revised version. See the

The following recommendation has been added.

Further research aimed at exploring perceived paternal fertility emotions across different cultural or socio-economic groups. Additional studies are needed which come up with a direct measurement of paternal emotional fertility intention.

---

## [Editor Report · Decision Letter 2]

21 Jan 2025

Perceived Paternal Emotional Fertility Intention and Its correlates in Ethiopia among a Cohort of Pregnant Women: Community based Longitudinal Survey; A Secondary Data Analysis of 2019/20 Baseline Survey

PONE-D-24-25872R2

Dear Dr. Solomon Abrha Damtew,

We’re pleased to inform you that your manuscript has been judged scientifically suitable for publication and will be formally accepted for publication once it meets all outstanding technical requirements.

Kind regards,

Yibeltal Alemu Bekele, MpH

Academic Editor

PLOS ONE

---

## [Editor Report · Acceptance letter]

PONE-D-24-25872R2

PLOS ONE

Dear Dr. Damtew,

I'm pleased to inform you that your manuscript has been deemed suitable for publication in PLOS ONE. Congratulations! Your manuscript is now being handed over to our production team.

Kind regards,

on behalf of

Mr. Yibeltal Alemu Bekele

Academic Editor

PLOS ONE